# Double Disparity of Sexual Minority Status and Rurality in Cardiometabolic Hospitalization Risk: A Secondary Analysis Using Linked Population-Based Data

**DOI:** 10.3390/healthcare11212854

**Published:** 2023-10-30

**Authors:** Neeru Gupta, Samuel R. Cookson

**Affiliations:** Department of Sociology, University of New Brunswick, Fredericton, NB E3B 5A3, Canada

**Keywords:** health services research, rural health, sexual minorities, healthcare inequalities, preventable hospitalization, data linkage, applied health statistics

## Abstract

Studies have shown separately that sexual minority populations generally experience poorer chronic health outcomes compared with those who identify as heterosexual, as do rural populations compared with urban dwellers. This Canadian national observational study explored healthcare patterns at the little-understood intersections of lesbian, gay, or bisexual (LGB) identity with residence in rural and remote communities, beyond chronic disease status. The secondary analysis applied logistic regressions on multiple linked datasets from representative health surveys, administrative hospital records, and a geocoded index of community remoteness to examine differences in the risk of potentially avoidable cardiometabolic-related hospitalization among adults of working age. Among those with an underlying cardiometabolic condition and residing in more rural and remote communities, a significantly higher hospitalization risk was found for LGB-identified persons compared with their heterosexual peers (odds ratio: 4.2; 95% confidence interval: 1.5–11.7), adjusting for sociodemographic characteristics, behavioral risk factors, and primary healthcare access. In models stratified by sex, the association remained significant among gay and bisexual men (5.6; CI: 1.3–24.4) but not among lesbian and bisexual women (3.5; CI: 0.9–13.6). More research is needed leveraging linkable datasets to better understand the complex and multiplicative influences of sexual minority status and rurality on cardiometabolic health to inform equity-enhancing preventive healthcare interventions.

## 1. Introduction

A growing volume of literature from different countries and regions demonstrates that sexual minority populations tend to experience disparities in sexual, mental, and behavioral health outcomes, some of which may be syndemically damaging to cardiometabolic health [1,2,3,4,5]. Such health disparities vary within and across sexual minority groups, which may include a diversity of persons who report any same-sex identity, attraction, or behavior [1,3]. For example, studies both in high-income countries [6,7] and in low- and middle-income countries [8] have found high levels of problematic substance use among women who have sex with women. In the United Kingdom, individuals who identify as lesbian, gay, and bisexual (LGB) have reported poorer markers of psychiatric morbidity than heterosexual individuals [9]. In Canada and the United States, some studies have indicated that LGB-identified adults experience higher rates of lifetime diabetes diagnosis than heterosexual patients, although diverging patterns have been seen across women and men [2,10]. Given the aging of the global population, the need to better understand differential chronic cardiometabolic disease risks and outcomes for sexual minority older adults is an increasing imperative [11].

The syndemics approach to understanding multiple drivers of cardiometabolic health inequalities has been grounded in the analysis that adverse outcomes are the result both directly and indirectly of heightened exposure to stressors from social stigma and discrimination associated with belonging to a sexual minority group, known as the minority stress model [12,13,14,15,16,17]. However, much of the evidence base on physical health conditions among sexual minority adults relies on data from small samples, self-reports, uncertain generalizability, and limited accounting for moderating variables [4,14]. It is increasingly argued that healthcare research on LGB populations should adopt novel methods, including integrating objectively verifiable data from routine medical records to reduce potential bias from self-reports [10,15].

Meanwhile, rural populations are widely shown to experience worse health outcomes than their urban peers, including adverse cardiometabolic outcomes and disproportionate premature mortality, likely associated with poorer access to primary care and other socioenvironmental inequalities [18,19,20,21]. It has been advanced that socioeconomic deprivation associated with rural residence may syndemically reinforce prolonged stress and, in turn, preventable and treatable chronic morbidity among minority populations [19,22,23]. Yet, many existing studies on rural health rely on small subsets of rural settings, limited accounting for individual socioeconomic status or socially entrenched gender norms, and approaches that fail to consider the heterogeneity of rural and remote communities [24,25,26,27]. The need for more research efforts focusing on long-term healthcare needs within diverse and evolving rural populations is increasingly recognized [21,28].

Little is known about cardiometabolic health outcomes at the intersection of disease progression, sexual minority status, and rural status. There has been a particular scarcity of healthcare research using representative samples distinguishing rural and urban LGB persons [29]. This quantitative observational study leverages multiple population-representative datasets integrating information on sexual identity, presence of diagnosed cardiometabolic diseases, potentially avoidable hospitalizations for severe cardiometabolic complications, rural and remote geographies, and sociodemographic characteristics of individuals in Canada. We aimed to address the following questions. (i) Do LGB adults have a higher risk of hospitalization for complications of diabetes and other cardiometabolic conditions than heterosexual adults? (ii) Is this risk moderated by degree of residential remoteness, unequal access to primary healthcare services, and other socioeconomic characteristics? (iii) Are the risks similar among women and men? The application entails a secondary analysis of a cohort of Canadian adults of working age with data on sexual identity, community rurality and accessibility, and other covariates of interest [30]. The Canadian context of universal publicly funded coverage of essential medical and hospital services for all residents should curtail financial barriers to primary and preventive care.

## 2. Methods and Materials

This study used linkable microdata files from three types of sources—population health surveys, routine hospital discharge records, and a geocoded remoteness index—to examine differences in hospitalization risk among sexual minority versus heterosexual adults of working age across rural and remote communities in Canada. General characteristics of each source can be found in Table 1, in line with international reporting guidelines for studies involving data linkage [31].

First, to obtain sufficient sample sizes of LGB-identified respondents, we pooled ten years of interview data from the 2008–2017 Canadian Community Health Survey (CCHS), a large-scale cross-sectional survey program implemented by Statistics Canada to help inform health policies and programs. The CCHS collects information annually on a number of health-related variables among individuals living in private households, designed to represent 97% of the total population [32]. The present analysis was limited to adults of working age (18–59 years) given differences in the salience of socioeconomic measures (e.g., employment status) and survivorship bias across the life span (notably in terms of survival prognosis after diagnosis), to those who provided consent to the national statistical agency to have their survey data linked with other administrative sources, and for whom valid data were available for self-reported sexual identity (heterosexual, lesbian or gay, or bisexual) and all other covariates of interest.

To obtain sufficient sample sizes of cardiometabolic-related hospitalizations, we then pooled ten years of administrative health data from the 2008/09–2017/18 Discharge Abstract Database (DAD), a national database updated annually (by fiscal year) on separations from acute care institutions in 12 of the country’s 13 jurisdictions (excluding the province of Quebec) [33]. Given Canada’s universal healthcare coverage system, these data are considered to offer an essentially complete recording of all inpatient stays for reporting jurisdictions. Following research approaches developed elsewhere [30], we considered as the outcome variable having at least one admission over the period of observation where the main diagnostic reason for the stay was for complications of selected cardiometabolic diseases: type 1 or type 2 diabetes mellitus, hypertension, cardiac arrhythmia, heart disease, heart failure, or stroke.

Third, we linked the CCHS and DAD datasets to the Index of Remoteness (IR), a continuous measure developed by Statistics Canada to gage all inhabited communities in terms of relative accessibility to services, road networks, and population centers [34]. For ease of interpretation, we classified the geographic-based index into quintiles (five mutually exclusive categories ranked according to IR value) to better understand the health implications of rural diversity within the context of Canada’s vast physical landscape [27]. In the absence of a standardized definition for the different degrees of rurality, generally perceived as being a social construct [25], we retained community quintiles 3–5 (i.e., the 60% highest remoteness values) as the more rural and remote parts of the country, that is, those characterized by more sparsely distributed populations and limited services and transportation infrastructures.

We conducted descriptive and multiple logistic regression analyses, stratified by sex (male or female, based on the available survey data). The multiple regression models were adjusted for residential remoteness as well as a number of person-level confounding factors in terms of sociodemographics (age, education, marital status, employment status), behavior-related risk factors (body mass index class, tobacco use, alcohol consumption), healthcare access (having a regular provider), and health status (reported lifetime diagnosis of diabetes, hypertension, heart disease, or effects of a stroke). Bootstrapped sampling weights were applied to the linked data to ensure the population representation of the results and robust 95% confidence intervals (CIs), assuming the characteristics over the period of observation reflected the combined average. All (unweighted) sample and (weighted) population counts were rounded and vetted to meet Statistics Canada data privacy and disclosure protocols.

## 3. Results

The cohort sample included 202,820 respondents ages 18–59 years with valid sexual identity information, representing 14,190,300 person-years of exposure to the risk of cardiometabolic-related hospitalization. As seen in Figure 1, based on the survey data, 2.8% of the working-age population identified as LGB and 97.2% as heterosexual. Most were currently working (80%), had at least some postsecondary education (71%), and were in a marital or common-law union (62%). In terms of health risks and status, 67% regularly consumed alcohol, 52% were living with overweight or obesity, 22% were current tobacco smokers, and 12% reported having been diagnosed with at least one chronic cardiometabolic condition (diabetes, heart disease, or hypertension, and/or effects of a stroke).

Significantly fewer LGB persons reported having a regular healthcare provider than heterosexual persons (80% versus 85%) (Table 2). This was seen despite Canada’s universal healthcare system, and despite significantly fewer LGB residing in the country’s more rural and remote areas (7% versus 10%). LGB persons more often exhibited modifiable health-related behaviors including tobacco smoking (31% versus 22%) and regular alcohol consumption (73% versus 66%). While LGB persons were found less often to have a diagnosed cardiometabolic condition, they did experience higher reported prevalence of heart disease compared with heterosexual persons (2.2% versus 1.8%).

Results of a multiple logistic regression among the whole study cohort did not reveal an independent association between LGB identification and risk of being hospitalized at least once for a cardiometabolic disease (*p* = 0.36), after controlling for primary healthcare access and other characteristics (not shown). However, when limiting the analysis to higher-risk adults (i.e., those having an underlying cardiometabolic condition) residing in more rural and remote communities, a different picture emerged. Findings illustrated a significant association of belonging to a sexual minority group with cardiometabolic-related hospitalization; LGB persons were approximately four times as likely to be hospitalized as heterosexual persons (odds ratio: 4.2; 95% CI: 1.5–11.7), adjusting for other factors (Table 3). As expected based on established epidemiological patterns, younger age and female sex were found to be protective factors in hospitalization risk. A certain increasing trend for degree of residential remoteness (IR quintile) with hospitalization risk was suggested, but the differences were not statistically discernible after adjusting for other characteristics.

Upon stratifying the analysis by sex, the association between sexual minority status and hospitalization risk remained significant among gay and bisexual men (adjusted odds ratio: 5.6; 95% CI: 1.3–24.4) (Table 4, model 2) but not among lesbian and bisexual women (3.5; 95% CI: 0.9–13.6) (Table 4, model 1).

## 4. Discussion and Conclusions

This study used multiple pooled and linked record-level and geographically based microdata sources in a secondary analysis of one of the largest known national cohort samples of women and men of working age with complete information on sexual identity (*n* = 5400 LGB-identified and 195,420 heterosexual-identified respondents aged 18–59) and a range of other sociodemographic, healthcare, and community characteristics [30]. The results contribute to an emerging evidence base on the complex intersections of sexual minority status, cardiometabolic health and risks, and rural status which may be exacerbating health inequities. We found that LGB persons reported to have a regular healthcare provider less often than heterosexual persons, while simultaneously residing less often in rural and underserved areas of Canada. Widespread rural barriers to healthcare, due to factors such as lack of medical practitioners and transportation problems, could worsen stigma-related health disparities among those who identify as a sexual minority; however, the evidence base is mixed, as LGB persons living in rural areas may also be more likely to report a greater sense of belonging than their urban peers [35]. Much of the existing literature on rural–urban differences in health outcomes is singularly limited to geographic barriers to healthcare access [26]; our analysis highlighted rural heterogeneity in access across socially marginalized groups. Results showed that LGB persons more often exhibited behaviors associated with chronic stress, including tobacco smoking and regular alcohol consumption, compared with heterosexual persons. They also experienced a higher prevalence of heart disease, albeit not diabetes. A previous study from the United States similarly reported diversity in the relationships between LGB identity and physical health, underlined by the tendency for research on LGB persons to focus more on indicators of mental health and wellbeing [35]. It has been theorized that sexual minority groups may be at heightened risk for cardiometabolic diseases stemming from exposure to stressors due to heterosexism, stigma, and prejudice, but supporting empirical evidence is fragmented and incomplete [12,15].

Our novel dataset linkage approach allowed us to examine how a range of social factors may independently contribute to potentially avoidable hospitalizations for complications of cardiometabolic disease, which entail substantial excess cost burdens for healthcare systems. Many studies using single (unlinked) sources do not distinguish health status as a contributor to hospitalization risk [36]. When limiting our analysis to adults with an underlying cardiometabolic condition and residing in more rural and remote communities, we found sexual minority status to be significantly associated with the higher risk of preventable cardiometabolic-related hospitalization; LGB persons were approximately four times as likely to have been hospitalized as heterosexual persons, after adjusting for age, sex, degree of community remoteness, and other characteristics. The pattern held as significant for gay and bisexual men when stratifying the analysis by sex, but not among lesbian and bisexual women. Studies elsewhere have found sex-stratified differences in lifetime diabetes diagnoses and other cardiometabolic health indicators by sexual identity, the causes and consequences of which remain largely unexplained to inform healthcare improvement [2,6,37,38]. For example, in a study from France, pregnancy-related modifiers of cardiovascular health risk were not found to differentiate outcomes between lesbian and heterosexual women, highlighting the need for more research on other potential biomedical and psychosocial mechanisms [38].

A limitation of this study was the exclusion of survey respondents without valid sexual identity information, as well as the lack of capture of other dimensions of sexual orientation (e.g., in terms of attractions or behaviors). While the proportion of nonresponses to the question on sexual identity was very small (0.97%), the characteristics of those who do not disclose such sensitive information according to predetermined categories may be different from those who do [30]. Although we did not find a significant association between degree of residential remoteness and hospitalization risk among those with an underlying cardiometabolic condition, statistical power was reduced when limiting the analysis to a small subgroup and for a relatively rare outcome. Moreover, the analysis was conducted for Canada, a high-income country characterized by universal healthcare coverage, considerable legal protections with regard to sexual minority populations [39], and a vast sparsely populated rural geography [26]. In countries with mixed healthcare payment systems, insurance status has been found to contribute significantly to health disparities among sexual minority groups, notably in the United States where the uninsured are numerous [31]. The generalizability of our findings to other contexts remains to be tested.

The present study reinforces the need for further research to support evidence-based clinical care to surmount stigma-related health disparities and meet the unique needs of sexual minority populations across different residential settings [35,40]. Studies elsewhere have pointed to a double disparity of sexual minority identity and rural residence on cardiometabolic risk factors such as tobacco use [23] and on interactions with healthcare providers [29]. At the same time, there is no universal delineation for how to identify in population-based datasets the continuum of sexual identities [41] or of rurality and remoteness [18,25]. This study explored the multiplicative effects on hospital-based cardiometabolic health outcomes from belonging at the intersection of two groups which often experience healthcare underservice but are typically examined separately in the literature.

## Figures and Tables

**Figure 1 healthcare-11-02854-f001:**
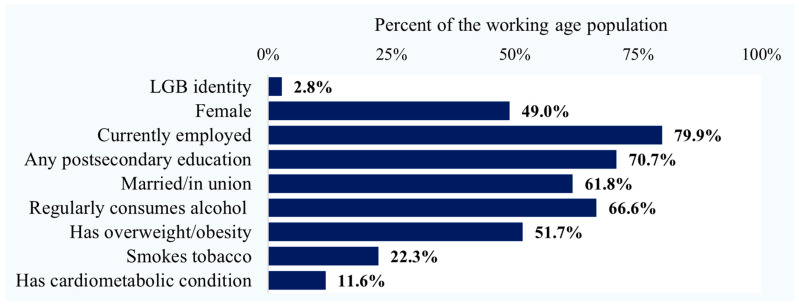
Percent of the population aged 18–59 by sexual minority identity, sociodemographic characteristics, and cardiometabolic health indicators.

**Table 1 healthcare-11-02854-t001:** Data sources, researcher-selected variables, and microdata linkage processes.

Domain	Dataset 1: Canadian Community Health Survey (CCHS)	Dataset 2: Discharge Abstract Database (DAD)	Dataset 3: Index of Remoteness (IR)
**Dataset coverage**	Household population ages 12 and over (10 provinces and 3 territories)	Hospital stays (9 provinces and 3 territories)	Geographically delineated communities (census subdivisions)
**Years included in the analysis**	2008–2017	2008/09–2017/18	2016
**Sample size (*n*)**	499,600 respondents	201,800 hospital stays	5125 inhabited communities
**Participants included in the analysis**	Adults ages 18–59 (*n* = 202,820)	Unique inpatients hospitalized for cardiometabolic diseases (*n* = 12,720)	60% most rural and remote communities (*n* = 3075)
**Records excluded from analysis**	Respondents outside working age range (*n* = 219,140), residence in province lacking DAD records (*n* = 59,880), nonresponse to sexual identity question (*n* = 1720), invalid person-level data on other covariates (*n* = 14,960), invalid geocode data (*n* = 3080)	Hospitalizations for all other conditions (*n* = 183,060), readmissions (*n* = 6020)	40% most urban and accessible communities (*n* = 2050)
**Linkage process**		Probabilistic matching by basic demographic characteristics (sex, birthdate, residential postal code)	Deterministic matching by residential postal code conversion to census subdivision

Note: Protocols for the CCHS-DAD linkage and Postal Code Conversion were developed by the national statistical agency; the present study used de-identified microdata files accessed in a secure university-based computing center. Source: Adapted from Gupta and Cookson [30].

**Table 2 healthcare-11-02854-t002:** Characteristics of the working age population (18–59 years) by sexual identity.

Characteristic	Lesbian, Gay, or Bisexual	Heterosexual
Has a regular healthcare provider	79.8% *	84.7%
Resides in a more rural or remote community	6.6% *	10.2%
Q3—less accessible areas	3.6% *	5.2%
Q4—remote areas	2.5% *	3.9%
Q5—very remote areas	0.5% *	1.1%
Regularly consumes alcohol	72.5% *	66.4%
Smokes tobacco	31.2% *	22.1%
Has overweight/obesity	45.9% *	51.8%
Has at least one cardiometabolic condition	9.3% *	11.7%
Diabetes (any type)	3.4%	3.9%
Hypertension	6.0% *	8.5%
Heart disease	2.2% *	1.8%
Effects of stroke	0.4%	0.4%
Hospitalized for complications of cardiometabolic disease	3.7% *	4.9%

Note: * = *p* < 0.05 (significantly different from the heterosexual group, based on Chi-square test). Residential remoteness based on quintiles of community accessibility/remoteness, with quintile 1 (Q1) = most urban/accessible areas of the country and quintile 5 (Q5) = most rural/remote areas. Source: Linked Canadian Community Health Survey, Discharge Abstract Database, and Index of Remoteness datasets (adapted from Gupta and Cookson [30]; *n* = 5400 LGB-identified and 195,420 heterosexual-identified respondents; data survey-weighted for population representation).

**Table 3 healthcare-11-02854-t003:** Results from the multiple logistic regression for risk of cardiometabolic-related hospitalization among adults aged 18–59 with an underlying chronic condition and residing in rural and remote communities.

Characteristic	Odds Ratio	95% Confidence Interval		*p*-Value
		Lower	Upper	
**Sexual self-identity**				
Lesbian, gay, or bisexual (LGB)	4.23 *	1.53	11.69	0.005
Heterosexual (ref)	1.00			
**Has regular healthcare provider**				
Has a regular provider	1.24	0.96	1.60	0.099
No (ref)	1.00			
**Interaction:** LGB identity * Has a regular healthcare provider	0.37	0.11	1.21	0.101
**Sex**				
Female	0.78 *	0.67	0.91	0.002
Male (ref)	1.00			
**Age group**				
Age 18–29 (ref)	1.00			
Age 30–44	1.45	0.92	2.27	0.108
Age 45–59	2.54 *	1.67	3.85	0.000
**Educational attainment**				
At most secondary (ref)	1.00			
Any postsecondary	0.95	0.81	1.10	0.479
**Body mass index class**				
Overweight or obesity	1.07	0.88	1.31	0.501
Not overweight/obese (ref)	1.00			
**Community remoteness (quintiles)**				
Q3—less accessible areas (ref)	1.00			
Q4—remote areas	1.13	0.96	1.32	0.144
Q5—very remote areas	1.15	0.93	1.43	0.193

Note: * = *p* < 0.05 [significantly different from the reference group (ref)]. Analysis among those with diagnosed diabetes, hypertension, heart disease, and/or effects of stroke. Model further adjusted for marital status, employment status, tobacco use, and alcohol consumption. Source: Linked Canadian Community Health Survey, Discharge Abstract Database, and Index of Remoteness datasets (authors’ calculations; *n* = 8420; data survey-weighted for population representation).

**Table 4 healthcare-11-02854-t004:** Results from the multiple logistic regressions for risk of cardiometabolic-related hospitalization among adults aged 18–59 with an underlying chronic condition and residing in rural and remote communities, according to sex.

Characteristic	(1) Female				(2) Male				
	Odds Ratio	95% Confidence Interval		*p*-Value	Odds Ratio		95% Confidence Interval		*p*-Value
		Lower	Upper				Lower	Upper	
**Sexual self-identity**									
Lesbian, gay, or bisexual (LGB)	3.45	0.87	13.60	0.077	5.63 *		1.30	24.39	0.021
Heterosexual (ref)	1.00				1.00				
**Has regular healthcare provider**									
Has a regular provider	1.03	0.72	1.47	0.859	1.37		0.97	1.93	0.077
No (ref)	1.00				1.00				
**Interaction:** LGB identity * Has a regular healthcare provider	0.59	0.12	2.86	0.516	0.20		0.04	1.07	0.060
**Age group**									
Age 18–29 (ref)	1.00				1.00				
Age 30–44	1.27	0.72	2.25	0.405	1.68		0.84	3.37	0.144
Age 45–59	1.84 *	1.08	3.12	0.024	3.52 *		1.85	6.70	0.000
**Body mass index class**									
Overweight or obesity	1.32 *	1.01	1.71	0.041	0.87		0.65	1.17	0.349
Not overweight/obese (ref)	1.00				1.00				

Note: * = *p* < 0.05 [significantly different from the reference group (ref)]. Analysis among those with diagnosed diabetes, hypertension, heart disease, and/or effects of stroke. Models further adjusted for employment status, tobacco use, and alcohol consumption. Source: Linked Canadian Community Health Survey, Discharge Abstract Database, and Index of Remoteness datasets (authors’ calculations; *n* = 4380 females and 4060 males; data survey-weighted for population representation).

## Data Availability

The microdata that support the findings of the study are available through Statistics Canada’s Research Data Centres, but restrictions apply to the public availability of these confidential data, which were used with permission for the current study.

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
