# Peer review of "Double Disparity of Sexual Minority Status and Rurality in Cardiometabolic Hospitalization Risk: A Secondary Analysis Using Linked Population-Based Data"

_healthcare, 2023, doi:10.3390/healthcare11212854_

Round 1
Reviewer 1 Report
Comments and Suggestions for Authors
Thank you for the opportunity to review the manuscript entitled, “Exploring the double disparity of sexual minority status and rural residence in cardiometabolic hospitalization risk: A secondary analysis using linked population based data” for possible publication in the journal Health Care. This manuscript reports on findings produced by linking multiple, high quality, health surveillance data sources used to describe risk for hospitalization from cariometabolic disease. Overall, I have a strong enthusiasm for this manuscript and its findings. However, there were a few issues, described below, that diminished my enthusiasm for recommending it for publication. With revision, I suspect this manuscript could make significant and important contributions to the existing evidence concerning health at the intersection of LGB identity and rural residence.
Abstract
· Please revise the language “heterosexual majority”. This language suggests that being in the majority is rationale for centering that identity and using it as a ‘normative’ referent group.
Introduction
· Page 1, line 37: Please revise to be consistent with the use of either “sexual minority” or “LGB” throughout the manuscript. If not sure which to choose, one could consider this publication (2020): https://nces.ed.gov/fcsm/pdf/fcsm_sogi_terminology_fy20_report_final.pdf
o Morgan, R.E., Dragon, C., et al. (2020). Updates in terminology of sexual orientation and gender identity survey measures, Federal Committee on Statistical Methodology.
· Page 2: A paper that may be useful in adding to the argument in the introduction, concerning the health of LGB folks in rural and urban areas, using health surveillance data sources, is the 2016 paper by Farmer et al.
o Farmer GW, Blosnich JR, Jabson JM, Matthews DD. Gay Acres: Sexual Orientation Differences in Health Indicators Among Rural and Nonrural Individuals. J Rural Health. 2016 Jun;32(3):321-31. doi: 10.1111/jrh.12161. Epub 2015 Dec 1. PMID: 26625172; PMCID: PMC4887433.
Discussion
· Generally the discussion lacks well-developed consideration of the findings in the existing literature pertaining to experiences had by LGB people in healthcare settings.
· Page 7, lines 207-211: The authors describe the ‘seemingly contradictory patterns’ in which LGB individuals more often reported not having a regular source of care AND less often endorsed living in rural areas. Authors could do a stronger job describing why this is a seemingly contradictory pattern in the context of the existing literature and what is documented about the experiences had by LGB folks in healthcare. 1 in 5 avoid healthcare setting due to fear of stigma, marginalization, and violence. LGB folks report experiences of microaggressions and macroaggressions when seeking healthcare. Therefore, it doesn’t seem strange or contradictory to see lower reports of having a regular source of care. It would be nice to see more consideration of the findings in the context of the empirically documented evidence about the experience of seeking care had by many LGB individuals.
· It makes sense that LGB persons who where high risk for cardiometabolic conditions where more likely to be hospitalized. For example, if individuals are fearful of experiencing negative outcomes, microaggressions, stigma, discrimination, etc in a clinical setting, they may be more likely to arrive at a clinical encounter with much later stage disease than someone who has a regular provider or who doesn’t have fear of stigma, discrimination etc. This could be more developed and considered in the discussion section.
· Limitations: It is also a limitation that only sexual orientation identity was used to describe sexual orientation. Sexual orientation is made up of identity, behavior, and attraction. Acknowledgement of how this measurement fact impacts the possible analyses and findings would be a meaningful addition to the limitations section.
· Limitations: Please address the fact that sexual orientation is a proxy for exposure to heterosexism. LGB sexual orientation does not cause the observed disparities, but the function of social systems, social norms, social contexts that occur across the social ecological model, cause the disparities.
Author Response
We thank the reviewer for their interest and thoughtful consideration of our submission. We have made a number of edits throughout, notable:
1) We have removed mention in the Abstract of the "heterosexual majority" (now referred in more neutral language as "those who identify as heterosexual", Line 10).
2) We have reframed the wording in the Introduction's first paragraph (lines 33-40), starting by including a definition for sexual minority groups which may refer to a diversity of identities, attractions, or behaviours (as supported by referencing, lines 34-35). Thereafter we tried to remain faithful to the language of the original source, e.g. where the research as used a "behaviour" measure among women, we retained the original neutral language of "women who have sex with women" (as cited on line 37). Where the cited research used an "identity" measure, we retained LGB-identified or similar language (lines 38-40). We have also reworded the variable labels in our original Tables 3-4 for consistency as appropriate.
3) We thank the reviewer for bringing the reference from Farmer et al. (2016) to our attention, which we have now cited in the Discussion (lines 205-208 and 257).
4) We acknowledge that we raised confusion in the Discussion by referring to ‘seemingly contradictory patterns’ - we have reframed our findings as: "We found that LGB persons reported to have a regular healthcare provider less often than heterosexual persons, while simultaneously residing less often in rural and underserved areas of Canada. Widespread rural barriers to healthcare, due to factors such as lack of medical practitioners and transportation problems, could worsen stigma-related health disparities among those identify as a sexual minority, although the evidence base is mixed..." (now lines 193-199).
5) The addition of an inclusive definition of sexual orientation upfront in the Introduction (per point 2 above) now flows to our mentioning later in the Discussion among the study limitations that this is a heterogeneous construct, but our source did not capture other dimensions of attractions or behaviours (lines 240-241).
6) We have explicitly mentioned the role of heterosexism as a stressor in the Discussion's first paragraph (supported by referencing, lines 209-211).
Reviewer 2 Report
Comments and Suggestions for Authors
The study by Gupta et al provides interestind findings about cardiometabolic health disparities among sexual minorities living in a rural area. The impact on cardiovascular health of these two recognized factors are known, but until now, little is known about cardiometabolic health outcomes at the intersection of disease progression, sexual minority status, and rural status. That’s what the authors’ study was designed to clarify.
The study is well built, well designed, based on strong data records, appropriately analyzed with regards on statistics, and also easy to read and to understand. The results are as follows :
- - Table 2 provides a relevant overview of the increased rate of CV risk factors among LGB minority patients (suitable for a broad readership interest)
- - Among LGB patients with an underlying cardiometabolic condition and residing in rural and remote communities, a >4-fold higher hospitalization risk was found for LGB-identified compared with their heterosexual peers.
- - In models stratified by sex, this association remained significant among gay and bisexual men but not among lesbian and bisexual women
However, the last result concerning differences between men and women should discussed taking into account two recently published studies showing apparently opposite conclusions:
- - Sexual minority status disparities in CV health, by O Deraz, et al, J Am Heart Assoc 2023 (12) e028429 : lesbian and bisexual women have a lower LE8 cardiovascular score compared with heterosexual women, but, conversely, homosexual and bisexual men have a higher LE8 CV score than heterosexual men
- - Ideal cardiovascular health in sexual minorities individuals, by BA Caceres, et al, JAMA Cardiol 2023 ; 8 : 335- : lesbian and bisexual women have a lower CVH score, and, conversely, homosexual and bisexual men have a higher CVH score
Author Response
We thank the reviewer for their thoughtful comments and appreciation of this work.
We are particularly appreciative of bringing to our attention the two new references on cardiovascular health (Deraz et al 2023; Caceres et al 2023). We have added and cited these relevant studies in the Discussion, particularly as informing venues for further research (Lines 235-238).
Reviewer 3 Report
Comments and Suggestions for Authors
The study is well-conducted with clearly defined objectives, data sources, and statistical methods the paper could provide a valuable contribution to the literature on healthcare disparities experienced by sexual minorities in rural settings. In my opinion it could be published as it is.
I provide some minor suggestions but the authors are free to implement it.
TITLE: The title is too long, make it shorter eg "Double Disparity: Sexual Minority and Rural Residence Impact on Cardiometabolic Hospitalization Risk in Canada."
Material and methods: Please explain why why quintiles 3-5 were specifically chosen as more rural and remote.
RESULTS
Line 151: The phrase "higher prevalence of heart disease" may be better phrased as "higher reported prevalence of heart disease."
Line 161: The phrase "all else being equal" is colloquial and could be replaced with "controlling for other factors" for clarity.
DISCUSION
The contradiction found could be explored further. “We found certain seemingly contradictory patterns, notably LGB 207 persons reporting to have a regular healthcare provider less often than heterosexual per-208 sons, while simultaneously residing less often in rural and underserved areas of the coun-209 try. “Why might this be the case? Any hypotheses or past research that might shed light on this? –
There's no discussion on how the omission of the province of Quebec from the DAD dataset (Line 102) might affect the results.
- Addressing differences among gay, bisexual men, lesbian, and bisexual women is important. However, stating that the causes and consequences "remain largely unexplained" is a bit vague. provide hypotheses or directions for future research to clarify. (Line 229-232)
The call for "further research to support evidence-based clinical care" is crucial. Suggestg specific areas of research or methodologies.
The challenge of identifying "sexual identities" and "rurality and remoteness" (Line 253-255)Suggest potential methodologies or standards to improve this in future studies.
Author Response
We thank the reviewer for the valuable feedback on our manuscript. We have made a number of revisions in response, as described below.
1) We appreciate the need to better balance the length of the Title with sufficient details for enhanced discoverability, notably given the thematic issue to which we are submitting ('Applied Statistics and Data Analysis in Healthcare'). We have reduced the title from 173 characters to 152: "Double disparity of sexual minority status and rurality in cardiometabolic hospitalization risk: A secondary analysis using linked population-based data."
2) In the Methods section, we have acknowledged that there is no standardized definition of rurality, and as such the choice of delineating quintiles 3-5 as more rural and remote was context-specific and socially-driven (Lines 122-123).
3) We have rephrased the two sentences: (1) from "they did experienced higher prevalence of heart disease" to "they did report higher prevalence of heart disease" (now Line 155); and (2) from "all else being equal" to "adjusting for other factors" (now Line 175).
4) We have reframed the Discussion on the "seemingly contradictory patterns" with better explanation and nuance, supported by new referencing (Lines 193-199; please also see the authors' response to the comments from Reviewer 1).
5) Regarding the "largely unexplained" sex-stratified differences mentioned in the Discussion, we have added more information for context and direction for further research, supported by a new reference, as follows: "For example, pregnancy-related modifiers of cardiovascular health risk were not found to differentiate outcomes between lesbian and heterosexual women according to one French study, highlighting the need for more research on other potential biomedical and psychosocial mechanisms (38)" (Lines 235-238).
6) Please note that, with regard to the present lack of hospital-based data from the province of Quebec, this is true for the multitudes of Canadian studies using the Discharge Abstract Database (DAD), which was originally developed in 1963. We are not aware of any studies evaluating potential impacts on national generalizability stemming from the separate reporting and analysis of hospital morbidity data from Quebec.